# A comparison of liver fat fraction measurement on MRI at 3T and 1.5T

**Lavanya Athithan** [1]*, **Gaurav S. Gulsin**[1], **Michael J. House**[2,3], **Wenjie Pang**[3], **Emer M. Brady**[1], **Joanne Wormleighton**[1], **Kelly S. Parke**[1], **Matthew Graham-Brown**[1], **Tim G. St. Pierre**[2], **Eylem Levelt**[1¤], **Gerry P. McCann**[1]

1 Department of Cardiovascular Sciences, University of Leicester and NIHR Leicester Cardiovascular Biomedical Research Centre, Glenfield Hospital, Leicester, United Kingdom, 2 Department of Physics, The University of Western Australia, Crawley, Western Australia, Australia, 3 Resonance Health Ltd, Burswood, Western Australia, Australia

¤ Current address: Multidisciplinary Cardiovascular Research Centre and Biomedical Imaging Science Department, Leeds Institute of Cardiovascular and Metabolic Medicine, University of Leeds, Leeds, United Kingdom

* la185@leicester.ac.uk

**Data Availability Statement:** Data cannot be shared publicly because of the nature of the human research participant data. The data contains potentially identifying or sensitive patient information and availability of this data would not

## Abstract

### Purpose

Volumetric liver fat fraction (VLFF) measurements were made using the HepaFat-Scan® technique at 1.5T and 3T to determine the agreement between the measurements obtained at the two fields.

### Methods

Sixty patients with type 2 diabetes (67% male, mean age 50.92 ± 6.56yrs) and thirty healthy volunteers (50% male, mean age 48.63 ± 6.32yrs) were scanned on 1.5T Aera and 3T Skyra (Siemens, Erlangen, Germany) MRI scanners on the same day using the HepaFat-Scan® gradient echo protocol with modification of echo times for 3T (TEs 2.38, 4.76, 7.14 ms at 1.5T and 1.2, 2.4, 3.6 ms at 3T). The 3T analyses were performed independently of the 1.5T analyses by a different analyst, blinded from the 1.5T results. Data were analysed for agreement and bias using Bland-Altman methods and intraclass correlation coefficients (ICC). A second cohort of 17 participants underwent interstudy repeatability assessment of VLFF measured by HepaFat-Scan® at 3T.

### Results

A small, but statistically significant mean bias of 0.48% was observed between 3T and 1.5T with 95% limits of agreement -2.2% to 3.2% VLFF. The ICC for agreement between field strengths was 0.983 (95% CI 0.972–0.989). In the repeatability cohort studied at 3T the repeatability coefficient was 4.2%. The ICC for agreement was 0.971 (95% CI 0.921–0.989).

be in compliance with the ethical approvals obtained. Data are available from the University of Leicester NIHR Leicester Biomedical Research Centre and the Leicester Clinical Trials Unit Institutional Data Access for researchers who meet the criteria for access to confidential data. Contact via Sally Utton, Research Manager Cardiovascular Imaging Research Group. All requests for data to be submitted in writing to su47@leicester.ac.uk.

**Funding:** The study was supported by the National Institute for Health Research (NIHR) Leicester Clinical Research facility and patients were recruited through a Career Development Fellowship granted to GPM. The NIHR provided support in the form of salary for GPM, research materials and study consumables. The salaries for LA and GSG was supported by Clinical Research Training Fellowships from the British Heart Foundation (BHF). The funders did not have any additional role in the study design, data collection and analysis, decision to publish, or preparation of the manuscript. The specific roles of these authors are articulated in the "author contributions" section.

**Competing interests:** MH is employed part time by Resonance Health Ltd; WP is employed by Resonance Health Ltd. MH holds shares in Resonance Health Ltd. TSP was a former consultant for Resonance Health Ltd. The authors would like to declare the following patents/patent applications associated with this research: TSP and MH are inventors on a patent (No. 2012350165) for measuring liver fat. HepaFat-Scan® is owned and marketed by Resonance Health Ltd. This does not alter our adherence to PLOS ONE policies on sharing data and materials.

## Conclusion

There is minimal bias and excellent agreement between the measures of VLFF using the HepaFat-Scan® at 1.5 and 3T. The test retest repeatability coefficient at 3T is comparable to the 95% limits of agreement between 1.5T and 3T suggesting that measurements can be made interchangeably between field strengths.

## Introduction

Non-alcoholic fatty liver disease (NAFLD) poses a significant healthcare burden with its incidence affecting 17–46% of adults in Western countries [1, 2]. A hallmark of NAFLD is the increased accumulation of triglyceride content within hepatocytes that results in steatosis [3, 4]. Many studies have shown that hepatic fat content is associated with obesity related metabolic complications [5, 6]. The global prevalence of NAFLD in type 2 diabetes (T2DM) is now 55.5% and the presence of insulin resistance and diabetes is considered a risk factor for more severe liver disease in NAFLD [7, 8].

Liver biopsy has been regarded as the gold standard to diagnose and stage NAFLD [1]. However, this is an invasive, uncomfortable procedure with significant procedural risks, including infection, and major haemorrhage [9]. Liver biopsy is also subject to sampling variability [10–12]. As such, appropriate non-invasive methods of liver fat measurements are desirable. Over recent years, proton density fat fraction (PDFF) has emerged as the preferred non-invasive quantitative imaging biomarker in the diagnosis and grading of hepatic steatosis [12, 13]. PDFF measurement by spectroscopy has been the accepted gold standard of reference used in the quantification of liver steatosis as it has the ability to measure the proton densities of triglyceride content within liver tissue [14–16].

PDFF measurements have been previously compared between 1.5T and 3T, using different manufacturers [16–20]. The largest comparison cohort across manufacturers was in n = 24 obese individuals which made a comparison of 1.5T Ingenia Philips, 3T Ingenia Philips and 3T 750 W GE. The mean Bland-Altman bias was -1.75% in the comparison of two Phillips scanner and -2.4% in the comparison of 1.5T Phillips against 3T GE [13, 18]. Yokoo et al. have conducted a meta-analysis that included 80 participants who had PDFF measured across both 1.5T and 3T using the same technique [16]. The mean bias associated with field strength for this meta-analysis was -1.2%. Other smaller cohorts have compared normal individuals against phantoms, diabetics against non-diabetics and children, with biases varying between -0.4 to +1.2% [17–20].

The fractional area of fatty vesicles seen in thin histopathology liver biopsy sections is numerically equivalent to the volumetric fraction of liver tissue occupied by fatty vesicles (the Delesse Principle) [21, 22]. HepaFat-Scan® is a proprietary magnitude-based MRI technique for measuring VLFF based on a series of 2D MR images that considers confounding factors. Adjustments are made to account for $T2^*$ decay, T1-amplification, noise bias, the differential between T1 relaxation times in water and fat, and the relative amount of MRI visible liver tissue. HepaFat-Scan® results have been shown to have negligible bias against biopsy and very high sensitivities and specificities for diagnosing all grades of liver steatosis [23]. The data acquisition protocol for HepaFat-Scan® was developed at a field strength of 1.5T, and has been modified for use at 3T prior to this study. The agreement between VLFF measurements acquired using HepaFat-Scan® measured at 3T and 1.5T has not been previously described. Furthermore, there have also been no previous studies comparing VLFF measurements of any

technique using two Siemens scanners at different field strengths. The comparative interstudy repeatability of the VLFF measurements using HepaFat-Scan® at the different field-strengths as well as at 3T alone has also not been defined.

The primary purpose of this study was to compare the agreement of VLFF measurements made using the HepaFat-Scan® technique acquired at 1.5T and 3T. We hypothesised that there would be good agreement and no significant bias between field strengths. We also assessed the interstudy repeatability of VLFF measurement using the HepaFat-Scan® technique at 3T.

## Methods

### Study design

This was a single-centre prospective, cross-sectional case-control study conducted at the National Institute for Health Research (NIHR) Leicester Biomedical Research Centre and University Hospitals of Leicester NHS Trust. The study was conducted with the approval of the West Midlands–Coventry & Warwickshire Research Ethics Committee (REC Ref: 15/WM/0222) and Solihull Research Ethics Committee (REC Ref: 17/WM/0192) in accordance with the ethical standards of the UK REC and Health Research Authority (HRA) in line with the Helsinki Declaration. All participants recruited were over the age of 18. Written informed consent was obtained on ethics committee approved consent forms that were stored in accordance with study protocol. Participants were also provided with a copy of the signed consent form.

### Study population

The baseline MRI scans of 90 participants from the DIASTOLIC trial (NCT02590822), assessing the effects of low-calorie diet and a structured program of exercise on cardiac structure and function were included for analysis. This consisted of 60 participants with diabetes and 30 healthy volunteers. Detailed study design and rationale as well as inclusion and exclusion criteria of participants are as previously described [24]. Healthy volunteers were recruited through interests from mailing out and poster advertisements. The inter-study repeatability cohort comprised of 17 participants from the PREDICT study (NCT03132129), made up of both diabetics and healthy volunteers.

### MRI scan protocol

Participants underwent serial MRI examinations that included a multiparametric cardiac scan as well as assessment of the liver and pancreas on a 1.5T (Aera, Siemens Medical Imaging, Erlangen, Germany) and 3T (Skyra, Siemens Medical Imaging, Erlangen, Germany) MR platform using an 18-channel phased array receiver coil. Both scans were conducted on the same day one after the other. The coil was centred over the participant's heart and liver. MRI data acquisition for measurement of VLFF via the HepaFat-Scan® protocol at 1.5T comprised an opposed-phase, in-phase, opposed-phase gradient echo sequence (TEs 2.38, 4.76, 7.14 ms, respectively, TR 88 ms, 1 excitation, flip angle 70 degrees, bandwidth 500 Hz) with a 10 second breath hold. Data from three 2D axial slices, positioned through the widest part of the liver, were acquired in a single breath-hold. The slice thickness was 4 mm and the matrix was 256 x 256 with a field of view 400 x 400 mm. The 3T HepaFat-Scan® acquisition used a similar gradient echo sequence again using three axial slices with modification of the TEs (1.2, 2.4, 3.6 ms, respectively), bandwidth 1300 Hz, and the number of slices. All other parameters were the same as the 1.5T acquisition.

## MRI image analysis

Quality control procedures were used to ensure data acquisition parameters included were accurate on all scans and conformed to the requirements of the protocol. Anonymised data were sent to Resonance Health for analysis and the HepaFat-Scan® result for both field strengths was calculated in an identical manner as previously described [23]. All analyses were conducted offline by two separate blinded observers at the core laboratory.

The measurements were processed by the HepaFat-Scan® software (Resonance Health Analysis Services Pty Ltd, Burswood, Australia) to generate a VLFF. Two regions of interest (ROI) about 580 mm$^2$ were delineated within the right and left lobes of the liver on each of the three MRI slices, avoiding large intrahepatic vessels and any obvious motion-affected regions. The image intensity was measured within the liver ROIs and also in an artefact-free region of free space outside of the patient to sample the background signal levels. Fig 1 illustrates a series of images that better describes this process. The background signal was subtracted in quadrature from the liver signal of each echo time before further processing. The raw T2*-corrected Dixon ratio, α, was calculated for each ROI as previously described [25]. To generate the

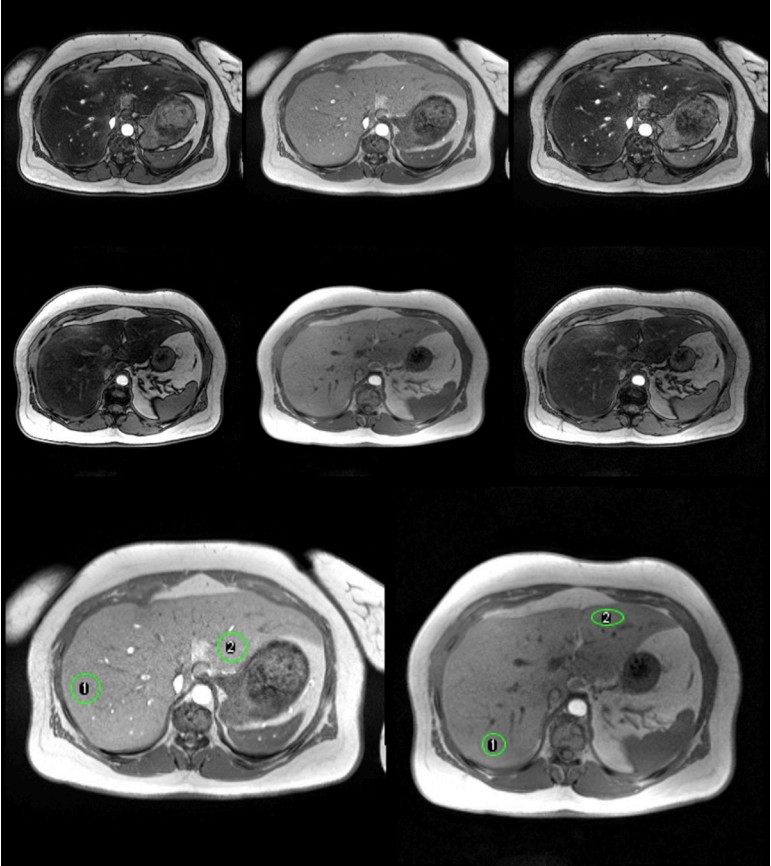

**Fig 1. Example of MRI image acquired and analysis process.** Top Row: Axial magnetic resonance images acquired at 1.5 T for VLFF measurement. Left: 2.38 ms (opposed phase), Middle: 4.76 ms (in phase), Right: 7.14 ms (opposed phase). Middle Row: Axial magnetic resonance images acquired at 3 T for VLFF measurement. Left: 1.21 ms (opposed phase), Middle: 2.38 ms (in phase), Right: 3.57 ms (opposed phase). Last Row: In-phase magnetic resonance images acquired at 1.5T (left) and 3T (right) for the same individual. The green ROIs indicate the two regions used for the analysis at each field strength. At 1.5T the VLFFs for regions 1 and 2 were 27.1% and 28.2%. At 3T the VLFFs for regions 1 and 2 were 28.5% and 28.0%.

VLFF, the α value is further corrected using the relationship defined between α, the VLFF and a constant k [25]. The constant k takes into account i) the specific sequence acquisition parameters (TR, flip angle), ii) the difference in T1 longitudinal relaxation times between fat and liver water, iii) the ratio of the proton density in fat to the proton density in non-fatty liver tissue, and iv) the volume ratio between the MR-visible water phase and the MR-invisible phase. The VLFF in each ROI was then averaged to produce a single unconfounded VLFF result.

The 3T analyses were performed independently of the 1.5T analyses. A different analyst, blinded from the 1.5T results, analysed the 3T data and no attempt was made to match the slices or regions of interest used in the 1.5T analyses.

### Inter-study repeatability

Interstudy repeatability of VLFF measured by HepaFat-Scan® at 3T was assessed in 17 participants (15 participants with T2DM and 2 healthy volunteers) completely independent of the previous cohort of 90 participants, consented to and completed a repeat scan at 3T under identical conditions within 20 days of their baseline visit. Acquisition parameters were identical to the first scan, but no attempt was made to match slice locations or analysis regions of interest and analysts were blinded to the identity of the participants.

### Statistical analysis

Data was analysed using SPSS Statistics version 25. Normally distributed data are shown as mean ± standard deviation and non-normally distributed data are expressed as median (interquartile range). The Bland-Altman method was used to calculate and display limits of agreement between measures and the presence of systematic bias between VLFF at different field strengths, and to assess the interstudy repeatability at 3T. The statistical significance of any bias was tested using a one-sample t-test. Two-way, random effect, intraclass correlation coefficients (ICCs) for absolute agreement was also used to assess the agreement of VLFF measures between repeat measurements at 3T.

## Results

### Baseline characteristics

The baseline MRI scans of 90 participants were included for analysis. This included 60 participants with type 2 diabetes and 30 healthy volunteers. Sixty seven percent of the participants with diabetes were male, and fifty percent of healthy volunteers were male. Mean age for participants with diabetes was 50.92 ± 6.56 yrs, mean BMI 36.50 ± 5.75 and for healthy volunteers mean age was 48.63 ± 6.32 yrs, BMI 24.32 ± 2.38 kg/m$^2$.

Mean VLFF values at both 1.5T and 3T were significantly higher in diabetics (12.65 ± 7.34% at 1.5 T;13.34 ± 7.77% at 3T) compared to healthy volunteers (2.72 ± 1.80% at 1.5T; 2.76 ± 1.68% at 3T).

### Interfield comparison and repeatability

Bland-Altman analysis showed good agreement and the mean bias between 3T and 1.5T was small at 0.48% (95% CI 0.19–0.77%) VLFF with 95% limits of agreement between 3.2 and -2.2% (Fig 2). This mean bias was statistically significant (t = 3.292, $P$ = 0.001).

There was excellent agreement between the two field strength measurements of liver VLFF with ICC = 0.983 (95% CI 0.972–0.989) for single measures. The descriptive statistics are detailed within Table 1 with graphical representation in Fig 2.

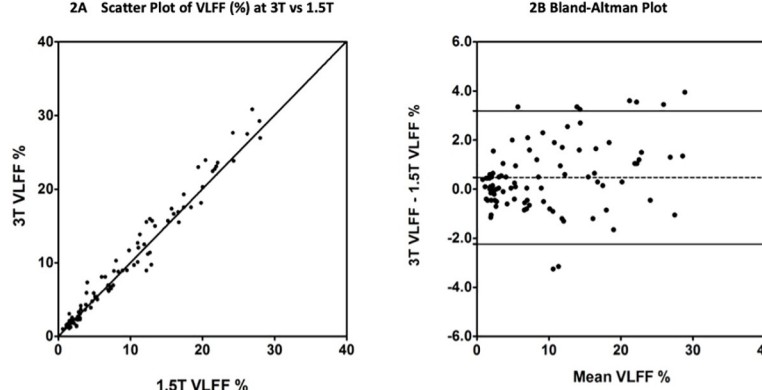

**Fig 2. Comparison of VLFF (%) at 3T vs 1.5T.** 2(A) Liver VLFF measurement at 3T plotted against Liver VLFF measurement at 1.5T for the 90 subjects. The solid line is the line of equivalence. 2(B) The difference between the VLFF measured at 3T and the VLFF measured at 1.5T plotted against the mean of the 1.5T and 3T measurements for the 90 subjects. The dashed line is the mean bias and the solid lines indicate +/- 95% limits of agreement.

### Test-retest repeatability

The group of subjects for repeatability testing consisted of 15 with diabetes and 2 healthy volunteers who were 53% male, mean age was 63 years and average BMI 29.8 kg/m$^2$. The average interval between scans for patients undergoing inter-study repeatability at 3T was 9.4 days (range 6–20 days). Table 1 details the significant statistical values for this analysis. The Mean VLFF % at 3T on visit 1 was 10.5 ± 8.99% and visit 2 10.9 ± 9.0%. Agreement was excellent with ICC of 0.971 (95% CI 0.921–0.989).

Fig 3 illustrates the VLFF repeatability data at 3T. The repeatability coefficient was 4.2% (95% CI 2.8–5.6%) at 3T indicating that 95% of pairs of measurements are expected to fall within 4.2% VLFF of each other. One pair of measurements from our study showed a large difference of 6% VLFF across a period of 7 days. While a real change in the VLFF cannot be ruled out, there were some technical differences between the data acquisitions. The first analysis from this participant was limited to a single ROI on a more inferiorly positioned slice compared to the second analysis where two ROIs (one in each lobe) were able to be positioned. Potentially this difference in anatomical position and number of analysed ROIs may have impacted on the repeatability of this case. Omitting this case would reduce the repeatability coefficient to 3.2%.

## Discussion

This is the largest single cohort of patients scanned on the same day across two field strengths on scanners of the same manufacturer to measure liver fat fraction. It is the first study to assess

**Table 1. Values and statistical tests for inter-field strength comparison and repeatability cohort.**

| *Interfield Comparison* | Values (95% CI) |
| --- | --- |
| Mean bias | 0.48 (0.19–0.77) % VLFF |
| 95% limits of agreement | -2.2 to 3.2% VLFF |
| ICC for absolute agreement | 0.983 (0.972–0.989) |
| *Repeatability Cohort* | Values |
| Repeatability Coefficient | 4.2 (2.8–5.6) % VLFF |
| ICC for absolute agreement | 0.971 (0.921–0.989) |

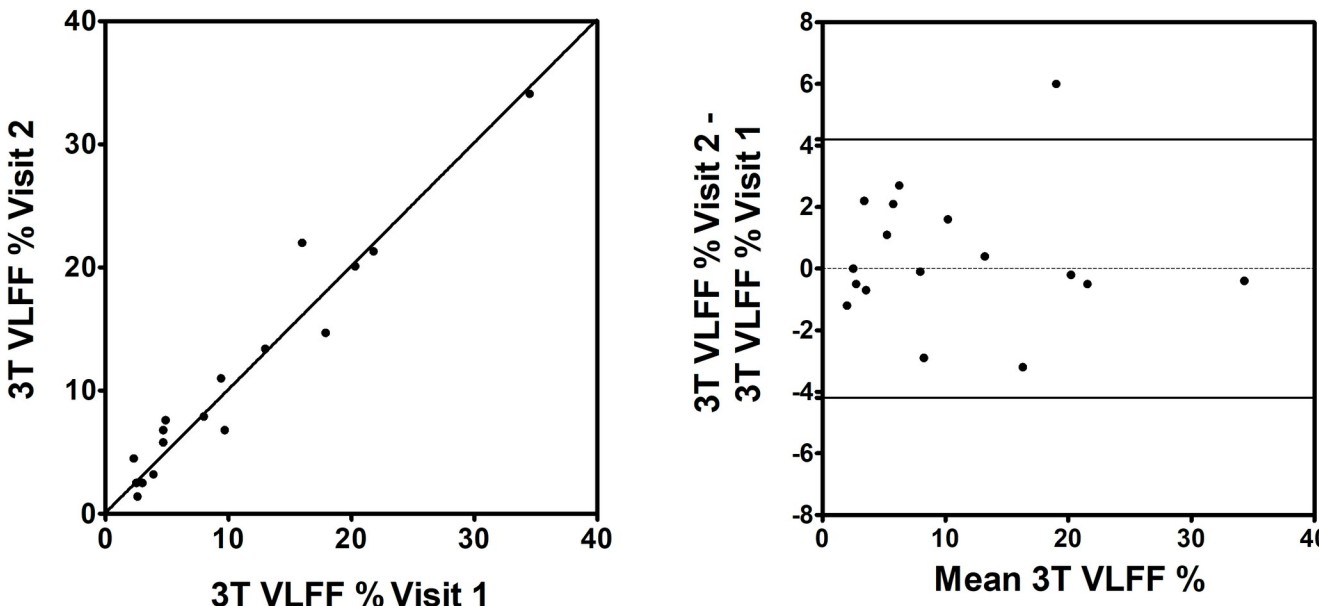

**Fig 3. Repeatability of VLFF Measurements at 3T.** (A) 3T VLFF at visit 1 plotted against 3T VLFF at visit 2. The solid line is the line of equivalence. (B) The difference between 3T VLFF at visit 1 and 3T VLFF at visit 2 plotted against the mean VLFF of visit 1 and visit 2. The solid lines indicate the 95% repeatability coefficient.

the degree of agreement of HepaFat-Scan® VLFF measurements at 3T with those at 1.5T. This study also includes a test-retest repeatability cohort. The measurements obtained span a large range and include both participants with diabetes as well as healthy volunteers. Even examined as individual groups, these groups are larger cohorts than have previously been studied in inter-field comparisons.

The data from this study indicate very good agreement between VLFF measures at these two field strengths. The small bias measured between 1.5T and 3T VLFF (0.48%, 95% CI 0.19–0.77%) is comparable with and generally smaller than other examples from the literature [13, 17, 18]. Although the mean bias was statistically significant, the small magnitude of the mean differences suggests that the interchangeable use of HepaFat-Scan results from 1.5T and 3T is unlikely to impact clinical decision making.

Although the VLFF values between the two groups are very different, with the participants with diabetes having strikingly higher values than healthy volunteers, the agreement remains excellent. This shows that at either field strength, HepaFat-Scan® VLFF measurements clearly distinguishes between health and disease states.

This study cohort has shown a positive bias in the difference between 3T and 1.5T MRI liver fat measurements (i.e. 3T VLFF higher on average than 1.5T VLFF). In the most recently published comparison of MRI-PDFF between two field strengths, both in GE scanners, although in a small population the magnitude of the bias was similar to our study, i.e. 0.4%, however it was a negative bias (i.e. 1.5T PDFF higher than 3T PDFF). Note that while PDFF and VLFF are different physical aspects of liver fat concentration, they are numerically very similar in magnitude and are strongly correlated with each other. Hence this is the basis of the validity of these comparisons. While two recent studies also reported that on average the MRI-PDFF at 1.5T was higher than at 3T, they also reported, conversely, that the 3T MRS-determined PDFF was higher on average than the 1.5 MRS-determined PDFF [13, 17]. In our

study, the 3T scan was always conducted after the 1.5 T scan and this could also be a contributing factor to the bias.

The test-retest measurement of Hepafat-Scan® at 3T is again novel and showed excellent repeatability suggesting precise measurements and robust analysis methodology. Our repeatability coefficient, 4.2 (± 1.4)% at 3T is higher compared to the 2.3 (± 0.3) % reported by Kim et al. [20]. A potential reason for this has been explained within the results section. Therefore, we have a technique that agrees at different field strength both in disease states and healthy volunteers and has good test-retest repeatability. Importantly, the repeatability coefficient is comparable to the 95% limits of agreement between the results from 1.5T and 3T. Taken together with the clinically insignificant bias between the results from the two field strengths, the data suggest that the HepaFat-Scan® method can be used interchangeably between the two field strengths for clinical purposes.

## Limitations

In principle, the small but systematic bias we measured could be used as a basis for correcting the 3T data, or vice versa, but as this is a single centre study confined to two specific Siemens scanners, it cannot be implied the results are applicable to other models and manufacturers although the excellent reproducibility is consistent with the previous literature. More data from other manufacturers would be required to determine whether the bias measured in this study using the Hepafat-Scan® methodology was applicable more widely. No phantom data acquisitions were made in this study, which may have limited the variability between field strength even further [26], however it is the measurement in patients that is clinically meaningful.

A limitation of this technique is that it does not sample the entire liver in 3D, but consists of three 2D slices. In the context of steatosis that is spatially heterogenous, this could present a potential problem in trying to colocalize images across longitudinal studies. Only one patient population was studied, diabetes, but this is unlikely to affect the results in other patients with steatosis since the analysis technique is generic.

## Conclusion

Inter-field strength agreement between the VLFF measured using Hepafat Scan® at 1.5 and 3T is good in a mixed cohort of subjects with diabetes and healthy volunteers. There was a systematic positive bias in VLFF measured at 3T compared to 1.5T but this difference was so small as to be considered clinically unimportant. Repeatability of VLFF measured using Hepafat Scan® at 3T was comparable to the 95% limits of agreement between 1.5T and 3T suggesting that measurements can be made interchangeably between field strengths.

## Author Contributions

**Conceptualization:** Michael J. House, Wenjie Pang, Tim G. St. Pierre, Eylem Levelt, Gerry P. McCann.

**Data curation:** Lavanya Athithan, Gaurav S. Gulsin, Joanne Wormleighton, Kelly S. Parke, Tim G. St. Pierre, Gerry P. McCann.

**Formal analysis:** Michael J. House, Wenjie Pang, Tim G. St. Pierre.

**Funding acquisition:** Eylem Levelt, Gerry P. McCann.

**Investigation:** Lavanya Athithan, Gaurav S. Gulsin, Eylem Levelt, Gerry P. McCann.

**Methodology:** Michael J. House, Wenjie Pang, Kelly S. Parke, Tim G. St. Pierre, Eylem Levelt, Gerry P. McCann.

**Project administration:** Lavanya Athithan, Gaurav S. Gulsin, Joanne Wormleighton, Kelly S. Parke, Gerry P. McCann.

**Resources:** Michael J. House, Wenjie Pang, Joanne Wormleighton, Gerry P. McCann.

**Software:** Michael J. House, Wenjie Pang, Tim G. St. Pierre.

**Supervision:** Michael J. House, Matthew Graham-Brown, Tim G. St. Pierre.

**Validation:** Lavanya Athithan, Michael J. House, Wenjie Pang, Tim G. St. Pierre.

**Visualization:** Wenjie Pang, Tim G. St. Pierre.

**Writing – original draft:** Lavanya Athithan.

**Writing – review & editing:** Lavanya Athithan, Gaurav S. Gulsin, Michael J. House, Emer M. Brady, Matthew Graham-Brown, Tim G. St. Pierre, Eylem Levelt, Gerry P. McCann.

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
