## [Decision Letter · Decision Letter 0]

23 Dec 2020

PONE-D-20-28831

A comparison of volumetric liver fat fraction measurement on MRI at 3T and 1.5T.

PLOS ONE

Dear Dr. Athithan,

Thank you for submitting your manuscript to PLOS ONE. After careful consideration, we feel that it has merit but does not fully meet PLOS ONE’s publication criteria as it currently stands. Therefore, we invite you to submit a revised version of the manuscript that addresses the points raised during the review process.

Although both reviewers thought this to be a well written manuscript, the scope was too narrow to make it very interesting. However, after major revision it still will be of sufficient interest, see the details listed by the reviewers.

We look forward to receiving your revised manuscript.

Kind regards,

Peter Lundberg

Academic Editor

PLOS ONE

Journal Requirements:

2.Please provide additional details regarding participant consent. In the ethics statement in the Methods and online submission information, please ensure that you have specified what type you obtained (for instance, written or verbal, and if verbal, how it was documented and witnessed). If your study included minors, state whether you obtained consent from parents or guardians. If the need for consent was waived by the ethics committee, please include this information.

3.Thank you for stating the following in the Competing Interests section:

"I have read the journal's policy and the authors of this manuscript have the following competing interests: Tim St Pierre holds shares in Resonance Health Ltd and consults to Resonance Health Ltd; Michael House holds shares in and is employed part time by Resonance Health Ltd; Wenjie Pang is employed by Resonance Health Ltd. Tim St Pierre and Michael House are inventors on a patent (No. 2012350165) for measuring liver fat. HepaFat-Scan® is owned and marketed by Resonance Health Ltd. The other authors have declared that no competing interests exist."

We note that one or more of the authors are employed by a commercial company: Resonance Health Ltd

c) We note that you have a patent relating to material pertinent to this article. Please provide an amended statement of Competing Interests to declare this patent (with details including name and number), along with any other relevant declarations relating to employment, consultancy, patents, products in development or modified products etc. Please confirm that this does not alter your adherence to all PLOS ONE policies on sharing data and materials, as detailed online in our guide for authors http://journals.plos.org/plosone/s/competing-interests by including the following statement: "This does not alter our adherence to  PLOS ONE policies on sharing data and materials.” If there are restrictions on sharing of data and/or materials, please state these. Please note that we cannot proceed with consideration of your article until this information has been declared.

Reviewers' comments:

Reviewer's Responses to Questions

**Comments to the Author**

1. Is the manuscript technically sound, and do the data support the conclusions?

Reviewer #1: Yes

Reviewer #2: Yes

2. Has the statistical analysis been performed appropriately and rigorously? 

Reviewer #1: Yes

Reviewer #2: No

3. Have the authors made all data underlying the findings in their manuscript fully available?

Reviewer #1: No

Reviewer #2: Yes

4. Is the manuscript presented in an intelligible fashion and written in standard English?

Reviewer #1: Yes

Reviewer #2: Yes

5. Review Comments to the Author

Reviewer #1: PLOS ONE PONE-D-20-28831 “A Comparison of Volumetric Liver Fat Fraction Measurement on MRI at 3T and 1.5T”

Summary: This is a well written article describing comparison of Volumetric Liver Fat Fraction (VLFF) aimed at determining the reproducibility of these measurements at 1.5T and 3T. The authors are using a proprietary technique, HepaFat-Scan, aimed at measuring liver fat content. As this is a proprietary technique, the technical details of this method are not well described and is a minor limitation of this work. The authors perform same day comparison in 60 patients and 30 healthy controls to assess the reproducibility of VLFF across field strength (1.5T v 3T). This has important practical consideration and therefore has value. The overall contribution to the field is modest, but these sorts of studies are important for those designing multicenter clinical trials where multiple MRI systems are being used. I have a number of comments that I believe should be addressed, as well as some larger philosophical issues that I describe below.

Specific Comments:

1. The term volumetric liver fat fraction is misleading. This implies that you are measuring volumetric liver fat content, ie: fat content over the entire volume of the liver. I realize that this is not what is being measured, but I would strongly discourage the use of this term, at least without a strong qualification early in the paper, and should be removed from the title of the paper. The term VLFF is unfortunate and arises from the calibration and assumption that the authors have performed in previous work comparing their method to the steatosis area fraction. The underlying assumption that volume fraction and area fraction from a biopsy slide are equivalent is not well accepted, in my opinion. It is certainly true that those will correlate well with one another, but I think that the term volumetric fraction is unfortunate and misleading. This term should either not be used in this paper or a major qualifier that addresses my concern is needed in the introduction and the abstract. I would ask that this term not be included in the title of the paper.

2. Related to the last point, a major limitation of this technique is that it is not “volumetric”. Rather, the method only measures liver fat over 3 slices. This is a major limitation for a variety of reasons including reproducibility for longitudinal studies. Unlike all commercial techniques that measure PDFF over the entire liver, ie: volumetric, the HepaFat-Scan technique does not. This is a problem for longitudinal reproducibility because it assumes that you can colocalize images across longitudinal studies, which is important because steatosis is known to be spatial heterogeneous. This limitation should be included in the Discussion. This is a same limitation that spectroscopy techniques have due to incomplete sampling of the liver (albeit not as bad for MRS).

3. This paper would have been much stronger had it done a direct comparison with VLFF and PDFF. Given that PDFF only requires a single breath hold for the entire liver, and these methods are available on Siemens scanners, I am a little surprised that this wasn’t performed. If these data are available, I would highly recommend that you include them here. It would be good to understand the relative performance across field strength of the two techniques.

4. Please specify that the HepaFat-Scan technique is a 2D technique. Please include the breath-hold time.

5. In the Introduction, please avoid the term “global pandemic”. Bad choice of words in 2020.

6. In the last paragraph of the first page of the introduction describing a cohort of 25 obese individuals, this study was performed only with GE scanners, not Philips and GE. Related to this general area the authors are making a big deal out of the fact that there are no other studies comparing the reproducibility of 1.5T versus 3T for liver fat measurements with MRI. This is somewhat disingenuous. It is true that there are not many studies with direct comparison on the same day, in the same subjects. However, if you read the Yokoo meta-analysis paper in Radiology which had over 1,600 patients, you will note that a large number of these patients has both 1.5T and 3T using the same spectroscopy technique. Therefore, that paper was able to assess the reproducibility of PDFF measurements across field strength in a very large number of patients. Please include those numbers in the introduction and please tone down the language highlighting the fact that this study has such a large number of patients to assess reproducibility. I do agree that the current paper is the largest study (60 patients, 30 healthy controls) with direct same day comparison, but the way it’s currently phrased is a little misleading.

7. Not many technical details about the proprietary technique are provided. This is fine but is a weakness of the study and should be acknowledged in the discussion. A few other questions that are remaining: how is the spectral complexity of fat considered in the fit for water and fat signals, and how are phase shifts related to concomitant gradients in eddy currents handled? Is this a magnitude-based technique? A few more details to this effect would be reasonable and helpful to the reviewers.

8. A limitation of the interstudy repeatability is that repeat scans 20 days after their baseline can lead to significant bias, especially if there is treatment that is occurring during this time (I believe this was one of the weight loss studies?). Further, variation in steatosis can occur over this time frame. The fact that 3T imaging occurred after 1.5T is another potential source of bias. These limitations should be addressed in the discussion.

9. First sentence of the discussion “largest single cohort of patients scanned on the same day across 2 field strengths”. Again, this is technically correct, but somewhat misleading considering the scope and magnitude of the Yokoo meta-analysis. Please see my comment above.

10. The figures appear very pixelated in my version. I am assuming that this was related to the submission process. Assuming that they are higher resolution, the quality of the figures looks good.

11. The use of a single vendor for this comparison limits the generalizability of this study. In addition, the Siemens scanners are actually 2.89T, not 3.0T, thus in my opinion, this sort of study would also need to be performed on Philips and GE systems in order to demonstrate generalizability. A comment to this effect, including the fact that the Siemens magnet is lower than 3.0T, should be included in the Discussion.

Summary: Overall, this sort of study has modest impact, but I do think is important for researchers looking to use the HepaFat-Scan method for multicenter studies that involve more than one platform, including 1.5T and 3T. I have a few minor concerns that I think should be easy for the authors to address. A limitation of this being a proprietary technique also limits my enthusiasm for the generalizability of these results. The limitation to a single vendor also limits my enthusiasm, as this does not provide the same of definitive reproducibility data that the Yokoo study provides for PDFF.

Reviewer #2: Summary

This study compares the measurement for liver fat fraction from two MR scanners from the same manufacturer at two different field strengths. There were two main research questions in this study. The first one was to evaluate the measurement of volumetric liver fat fraction (VLFF) when data has been acquired on a 1.5 T and a 3.0 T scanner. Both scanners were Siemens scanners and the data was acquired the same day. The second research question was to evaluate the repeatability of the VLFF measurement at 3T. They conclude that the repeatability results from the two 3T examinations are relatable with the 95 % limits of agreements of the 1.5 T and 3 T comparison, in which a significant but small bias was observed. Hence, they claim that the method for VLFF could be used interchangeable from choice of MR-scanners.

The authors present the study in a clear, easy interpretable way. It is easy to follow, clearly written and the authors addresses the literature well regarding alternative methodology for fat fractions measurement (although not comparing why an alternative to the PDFF measurement is needed and what benefits VLFF might addresses). However, in order to interpret the result properly and interpret the authors conclusions I do miss a few analyzes in the result part, mainly confidence intervals for the ICC as well as a residual plot/regression line for figure 2 and figure 3 to investigate potential proportional biases. My recommendation for this paper is therefore Minor Revision.

Below follows some discussion regarding specific areas for improvement:

Major issues

1. Introduction (Page 4, Row 53). At page 3 other common techniques are discussed for fat estimation using invasive a non-invasive techniques are discussed. However, page 4 starts with introducing VLFF without comparing it to the methodologies on page 3 more than it has comparable results to biopsy (based on ref 21). I miss a small motivation for VLFF as a technique. What benefits do it have compared to invasive methods or to spectroscopy and PDFF? Or are they totally interchangeable? Is there any situations where VLFF could be more beneficial?

2. Method (Page 6, Row 107): The number of slices for the 3 T acquisition is missing.

3. Method (Page 7, Row 131): The algorithm addresses correction for confounding factors including correction for different T1 relaxations times between fat and water signal, T2*-correction etc. However, no remarks are adressessed regarding B0 and B1 inhomogeneity. Is the VLFF measurement sensitive to that and how would it affect the results?

4. Method (Page 7, Row 151): Were there any reason for the “not on the same day” of the inter-scanner repeatability investigation at the 3T scanner? Would there be any benefits on testing the variability on the same day? `

5. Results (Page 9, Row 186): Please include the confidence interval (CI) for the ICC calculations (both for 1.5 vs 3T comparison as well as the 3 T repeatability investigation. Rephrase (if needed) the statement of the agreement. Include the CI in Table 1 and abstract.

6. Results (Page 9, Row 189) Table 1: Change “Values (CI)” change to “(95 % CI)”. Indicate the significance of the bias would help the interpretation of table 1 without needing to read the main text. Table 1 would increase the possibility to interpret the results if indicating the significant bias.

7. Results (Page 9, Row 176; Page 25 (Figure2)) No Residual plot or any analysis is done to investigate a potential proportional bias of the correlation. An overall overestimation of 3T compared to 1.5 T is observed. Although the bias is small, it might be valuable to investigate any proportional biases if intended use is for diagnosing fatty liver. Especially since the cut-off between healthy and fatty liver is considered to be as low as 5 %. Is it therefore possible to include a report of the looking at the residual plot or add a regression line at Figure 2 (and 3)?

8. Results (Page 25) Figure 2: Is it possible to increase the resolution of the Figure to increase readability? (In the PDF link I read and when I downloaded the .tiff image I found it hard to read figure 2 and figure 3. A higher resolution would increase the readability a lot. )

9. Results (Page 26) Figure 3: Is it possible to increase the resolution of the Figure to increase readability? (In the PDF link I read and when I downloaded the .tiff image I found it hard to read figure 2 and figure 3. A higher resolution would increase the readability a lot. )

10. Discussion (Page 11, Row 233): I find the paragraph starting with “It is also interesting to note that although the VLFF values between the two groups are very different…” is a bit out of scope, not necessarily correct, and hard for the reader to interpret themselves based on what is presented in the result section. The authors claim the VLFF clearly distinguish between health and diseased states but the only measurement we have access to as a reader is the mean value and standard deviation. So based on the population level, it seems that patients diagnosed with type II diabetes seems to have a higher fat fraction in there liver but we don't know the overlap or the distribution of the distribution of the fat content. However, the cohort are not included based on their known fatty liver status, but if they have a type II diabetes diagnose. I suggest removing this paragraph alternatively re-phrasing it to avoid misunderstandings. I don’t think the conclusions from that paragraph is possible based on the data that was provided.

Minor issues

11. Results (Page 24) Figure 1: Is it possible to increase the size of number 1 and 2 of the ROI:s alternatively increase the resolution a bit for readability?

12. Results (Page 25 - Page 26) Figure 2 and Figure 3: In Figure 2, Bland-Altman is written with an “-“ in the middle. In Figure 3, Bland-Altman is written as to separate words. Consider being consequent.

13. Results (Page 26) Figure 3: In figure 3, indicate the unit (%) on the axes.

6. PLOS authors have the option to publish the peer review history of their article (what does this mean?). If published, this will include your full peer review and any attached files.

Reviewer #1: No

Reviewer #2: No

---

## [Author Response · Author response to Decision Letter 0]

6 May 2021

Detailed response to each comment within "Response to Reviewer" document.

System will not allow all comments to be posted here despite multiple tries. ? Too large

---

## [Editor Report · Decision Letter 1]

26 May 2021

A comparison of liver fat fraction measurement on MRI at 3T and 1.5T.

PONE-D-20-28831R1

Dear Dr. Athithan,

We’re pleased to inform you that your manuscript has been judged scientifically suitable for publication and will be formally accepted for publication once it meets all outstanding technical requirements.

Kind regards,

Peter Lundberg

Academic Editor

PLOS ONE
---

## [Editor Report · Acceptance letter]

5 Jul 2021

PONE-D-20-28831R1 

A comparison of liver fat fraction measurement on MRI at 3T and 1.5T 

Dear Dr. Athithan:

I'm pleased to inform you that your manuscript has been deemed suitable for publication in PLOS ONE. Congratulations! Your manuscript is now with our production department. 

Kind regards, 

on behalf of

Professor Peter Lundberg 

Academic Editor

PLOS ONE